# Violence Under Control: Self-Control and Psychopathy in Women Convicted of Violent Crimes

**DOI:** 10.3390/bs15050656

**Published:** 2025-05-12

**Authors:** Emma De Thouars Da Silva, Sofia Knittel, Afonso Borja Santos, Bárbara Pereira, Andreia de Castro Rodrigues

**Affiliations:** William James Center for Research, Ispa—Instituto Universitário, 1149-041 Lisboa, Portugalarodrigues@ispa.pt (A.d.C.R.)

**Keywords:** female criminality, self-control, psychopathy, substance use, violent crimes

## Abstract

Despite the increase in the study of women and crimes committed by them, investigations continue to be scarce. Self-control and psychopathy have been widely studied in incarcerated populations, though more frequently in males than females. This study examines these psychological variables related to substance use history and violent crime in a sample of 94 incarcerated women in Portugal. Participants completed a sociodemographic questionnaire, the Self-Control Scale, and Levenson’s Self-Report Psychopathy Scale—VP. We found average self-control levels, with lower scores among participants with substance use. Significant differences in self-control emerged between women who committed homicide and those who did not. Psychopathy scores were above average, with significant differences in Factor 2 (impulsivity) between those who committed homicide and those who did not. Self-control and psychopathy were negatively associated, and psychopathy predicted self-control. These findings, which are not entirely consistent with the literature, challenge common assumptions about self-control, psychopathy, and crime, particularly in incarcerated women, and suggest that different mechanisms may drive violent and non-violent crimes in women. These results reinforce the need to consider gender-specific pathways to crime, highlighting the urgency of continuing to investigate the manifestation, in women, of widely studied variables in male samples.

## 1. Introduction

Since the 1980s, the study of women and female criminality has experienced an exponential increase ([4]). However, this research continues to have a generally scarce presence ([11]), and therefore, it is crucial to continue investigating in order to address certain issues that (still) remain very relevant, like the manifestation, in women, of certain clinical constructs widely studied in men, such as self-control and psychopathy ([82]; [95]).

Globally, it is estimated that the prison population exceeds 10.7 million individuals, including both individuals in pretrial detention and serving a prison sentence ([35]). Specifically, female crime is defined and characterized by its lower occurrence when compared to men ([64]); however, there has been an increase in the female incarceration rate ([91]). It is estimated that in August 2022, more than 740,000 women and girls were incarcerated, either convicted or in pretrial detention, representing an increase of about 60% in the number of incarcerated women and girls since the year 2000 ([34]). In Portugal, female incarceration presents some particularities. Since the 2000s, there has been a significant change in the number of imprisoned women in Portugal, with 1216 women incarcerated in 2000, 875 in 2005, and 627 in 2010 ([34]). The decriminalization of psychoactive substance use in 2001 may be the reason for this change ([14]), as it resulted in the reduction in the detention rate of women ([21]). Nevertheless, recent years, namely after the decriminalization of drug use, have been characterized by an increase in the female prison population, with 796 incarcerated women in Portuguese prisons as of 31 December 2020, and 906 women at the end of 2023 ([30]). Moreover, as of 31 December 2023, there were 9538 convicted individuals in Portugal, of whom 7.1% (673) were women ([30]), making it essential to thoroughly explore this issue and its specificities, since historically, criminal justice models ([9]), support systems, and post-sentencing programs are primarily based on male needs and experiences ([3]).

### 1.1. Self-Control

The human being resists, daily, a series of impulses and desires that are pleasurable in the moment but may have negative long-term consequences and violate what are considered appropriate social norms ([5]). Thus, self-control concerns the ability of an individual to nullify their impulses and behave according to established social norms. Deficits in this construct have been strongly associated with crime, violence, and conduct problems ([89]).

According to the General Theory of Crime, one of the main explanatory perspectives of crime, at the root of antisocial behaviors, including crimes, is low self-control. This personality characteristic controls an individual’s ability to resist opportunities for transgression, another factor intrinsically related to crime commission according to the GTC ([41]). Thus, and contrary to popular belief, most criminal acts do not involve a deep and careful consideration of the risks and benefits of committing the crime, but rather a failure in self-control([28]). Accordingly, it is argued that offenders make the decision to transgress based on immediate consequences and benefits, not considering long-term outcomes. Therefore, individuals with higher self-control tend to have a deeper perception of the long-term consequences of their actions and a higher chance not to transgress ([41]). Behaviors such as excessive alcohol consumption, the use of psychoactive substances, and smoking are considered, by the authors of the GTC, to be reckless behaviors undertaken by individuals with low levels of self-control ([41]). A study that aimed to further investigate the relationship between self-control and various forms of substance use found that, in a sample of university students, there was a greater tendency to report marijuana use, excessive alcohol consumption, and the inappropriate use of prescription medication ([37]). Additionally, a study conducted with adolescents concluded that the lower the levels of self-control, the greater the propensity for tobacco, alcohol, and marijuana consumption ([27]).

Certain specific forensic populations have been linked to low levels of self-control, such as individuals in prison ([59]), and, in this sense, other studies also show that there is an association between low levels of self-control and violent crimes ([76]), such as homicide ([33]). Notwithstanding all that has been mentioned, it is essential also to understand that “lack of self-control does not necessarily lead to crime, as it may be counterbalanced by circumstances and other individual characteristics” (p. 89) ([41]).

Additionally, research suggests that women generally exhibit higher levels of self-control than men ([75]; [81]), which may contribute to their lower involvement in criminal activity ([83]). This can also be understood through the lens of informal social control. Historically, women have been subjected to stricter and more intense informal social control compared to men ([54]), which has clear implications for female transgression ([62]). Consequently, this heightened social control may help account for the lower number of women in the prison system, whether in pretrial detention or serving sentences ([22]). [41] ([41]) also admit, in their work, the existence of differences between the sexes, arguing that men have lower levels of self-control than women. However, since most of the literature concerning self-control focuses on men who transgress ([9]; [80]), we highlight the importance of developing and investigating this variable and its manifestation in women.

### 1.2. Psychopathy

Scientifically known as antisocial personality disorder, as it is described in the DSM-5 ([1]), psychopathy is characterized by a universal pattern of disregard and disrespect for others, which starts to develop in childhood or adolescence and persists into adulthood. However, to confirm this diagnosis, the individual must be at least 18 years old. Psychopathy is a clinical construct characterized by irresponsibility, difficulty adjusting to social norms, high impulsivity, inability to plan for the future, aggressiveness, and also a lack of remorse and empathy ([1]).

Undoubtedly, the creation of the construct of psychopathy represents one of the most fundamental developments in the field of forensic psychology ([69]). As a personality disorder, it is one of the most studied by the scientific community due to the (possible) impact caused by the behaviors associated with the disorder, which are often related to criminal and delinquent behavior ([87]).

It is important to note, however, that criminal behavior does not only occur in individuals with this disorder. Similarly, there is no causal link between individuals diagnosed with antisocial personality disorder and the commission of illicit acts. Furthermore, at the moment of the adoption of criminal behavior, it is necessary to consider the effect of other variables beyond psychopathy ([70]).

It is suggested that individuals with these traits tend to focus on their interests and motivations, acting to obtain them, and manipulating others without any remorse or empathy ([24]). In line with this, it is worth noting that this construct is structured around two factors: Factor 1, referred to as the Interpersonal/Affective dimension, encompasses personality traits such as superficiality, manipulation, and lack of guilt, remorse, and affect; while Factor 2 is more commonly associated with an antisocial and self-destructive lifestyle ([16]; [93]). Historically, the construct of psychopathy and its diagnosis did not carry significant value in predicting and understanding criminal behavior. Only in recent years, with the changing role of psychopathy in the criminal justice system, has it become clear that individuals with psychopathic personalities are much more likely to transgress laws and socially established norms ([44]). Congruently, individuals diagnosed with psychopathy are believed to be more likely to commit the crime of homicide since their motivations and goals drive them and, consequently, may behave more impulsively, violently, and aggressively ([72]).

It is estimated that the prevalence of this disorder in the general population is approximately 1%, while in the prison population, it is around 25% ([23]), which underscores the critical need for targeted psychological intervention in the prison context.

The study of this construct has mostly been conducted with males, with scarce research on psychopathy regarding its existence and manifestation in females ([95]). Over the years, male-centric criteria have been applied to women, overlooking gender differences and the role of gender ([38]), increasing the challenges faced by researchers and mental health professionals in evaluating psychopathy in women, often leading to this personality type being misidentified as other disorders (e.g., borderline personality disorder) ([67]). Additionally, there are clinical differences in antisocial behaviors when compared to males, which further complicates the identification of this disorder in women, suggesting that some women may go undiagnosed ([40]). The increased difficulty in diagnosis inevitably leads to less effective interventions, particularly for women who have committed crimes or who are violent ([67]).

In line with what has been previously mentioned, this disorder manifests itself in both men and women but with distinct particularities, especially regarding behavior ([40]; [90]), the form and severity of the practiced violence ([18]), and a higher tendency, by women, for the use of psychoactive substances ([13]; [60]). Thus, given that there are, in fact, differences between men and women in the manifestation of psychopathy, the criteria considered as indicators of psychopathology in men may not be the most appropriate when identifying it in women ([38]).

Furthermore, it is believed that the prevalence of psychopathy is much lower in female samples compared to male samples ([18]; [31]). However, the number of studies associating the variable of psychopathy specifically with females is also significantly lower when compared to that of males ([40]; [57]). The gender variable is often overlooked in forensic developments, resulting in constructs, such as psychopathy and the instruments associated with it, for female samples, remaining vague ([31]), thereby emphasizing the relevance of this study.

Finally, self-control and psychopathy are considered explanatory theories of antisocial behavior and share some similarities, in the sense that both constructs are evident in individuals who have a tendency to be more impulsive rather than self-regulated, and, as opposed to having a cognitively and prudently action-oriented approach, are much more prone to risky actions ([25]). Both variables, despite being notable explanatory theories of antisocial behavior and although the characteristics of each result in very similar individuals, are insufficiently articulated and studied together ([26]), especially regarding women that committed crimes.

### 1.3. The Present Study

Compared to other European countries, Portugal has a notably high rate of female incarceration ([34]). However, research on women who commit crimes within the Portuguese context remains scarce ([63]). As mentioned throughout the Introduction, constructs such as self-control and psychopathy are also more frequently studied in male forensic populations than in female ones, and likewise, the relationship between these constructs is equally crucial yet underexplored ([26]). Thus, to contribute to this line of research, the present study aims to explore self-control and psychopathy in a population of incarcerated women.

Thus, this study hypothesizes that there are differences in self-control and psychopathy levels among women depending on their engagement in specific behaviors. First, we propose that women who use psychoactive substances will show differences in both self-control (H1) and psychopathy (H2) levels compared to those who do not. Additionally, this study explores whether women who have committed violent crimes, such as homicide and robbery, exhibit differences in self-control (H3) and psychopathy (H4) compared to women who have not committed these crimes. Finally, we hypothesize that there is a significant relationship between self-control, psychopathy, and the commission of violent crimes among convicted women (H5).

## 2. Methodology

### 2.1. Participants

This study included 94 incarcerated women in Portugal who met the following inclusion criteria: (a) understanding and reading the Portuguese language, (b) being over 18 years of age and, (c) serving a prison sentence (convicted). That being said, the sampling method used in this study was non-randomized by convenience.

### 2.2. Instruments

#### 2.2.1. Sociodemographic Questionnaire

The sociodemographic questionnaire aimed primarily at collecting specific information about certain sociodemographic variables of the participants, allowing for a better understanding of the characteristics related to participants’ incarceration.

#### 2.2.2. Levenson’s Self-Report Psychopathy Scale—VP (LSRP-VP) ([17])

Since one of the objectives of this study is to investigate the presence of psychopathic traits in participants, we chose to use the LSRP-VP, which was adapted and validated for the Portuguese population by Coelho, Paixão, and Silva in 2010. This instrument consists of a total of 26 self-report items on a Likert scale ranging from 1 to 4. The instrument has a bifactorial structure: Factor 1 (F1) is associated with primary psychopathy and Factor 2 (F2) with secondary psychopathy. The primary psychopathy scale comprises 16 items and refers to traits such as the manipulation of others and selfishness, while the second scale (secondary psychopathy) consists of 10 items and includes aspects such as a self-destructive lifestyle and impulsivity ([16]).

#### 2.2.3. Self-Control Scale ([19])

Originally developed by [88] ([88]) and translated and adapted for the Portuguese population by [19] ([19]), the Brief Self-Control Scale consists of a total of 13 items, answered on a Likert scale ranging from 1 (“not at all”) to 5. (“very much”) The total score for the scale can range from 13 to 65 points, with higher scores indicating higher levels of self-control, with a mean of 42.75 and an SD of 6.87 ([19]).

### 2.3. Procedures

We began by contacting the authors of the instruments—the 20-Item Social Desirability Scale ([84]), the Self-Control Scale ([19]), and the LSRP-VP ([17])—to request authorization for their use in the research. Simultaneously, authorization was requested from the Directorate-General of Reintegration and Prison Services (DGRSP) to conduct data collection in the selected prison facilities. Once permission was granted, we contacted the prisons to schedule data collection sessions and initiate the administration of the questionnaires.

The data collection took place between April 2023 and August of the same year. The questionnaire administration always began with participants reading and signing an informed consent form, thus ensuring confidentiality and anonymity. Additionally, participants were informed of their right to withdraw at any time without any consequences and of the voluntary nature of their participation. The completion of the instruments took approximately 30 min.

### 2.4. Data Analysis

The statistical analysis of the collected data was performed using the IBM SPSS software—Statistical Package for the Social Sciences—version 29.0.0.0.

Through descriptive analyses, we were able to describe and characterize the sample, as well as report the mean values for self-control and psychopathy. Before testing the aforementioned hypotheses, we first verified the assumptions of normality and homogeneity of variances. Using the Kolmogorov–Smirnov test, we determined that the distributions of the variables did not follow normality (*p* < 0.05). Consequently, non-parametric tests were used throughout the study.

To assess differences in self-control and psychopathy between participants with and without substance use, a Mann–Whitney U test was conducted.

We opted to explore the variables related to violent crimes, namely homicide and robbery, as only three participants were convicted of domestic violence, and no participants were convicted of other violent crimes in our sample. To investigate differences in the variables between participants who had committed homicide and those who had not, as well as between those who had committed robbery and those who had not, we also conducted the Mann–Whitney U test. However, it is worth noting that the data on these types of crimes are merely quantitative (i.e., having committed it or not), and no information was provided to us in terms of the circumstances or motivations of the crime commission.

To determine whether there was a significant correlation between self-control, psychopathy, and the commission of violent crimes, a Point-Biserial Correlation was performed, given the presence of a dichotomous nominal variable.

Finally, based on the results of this correlation, we then performed a simple linear regression between self-control and psychopathy to examine whether there was a predictive effect between the variables.

The level of statistical significance used throughout the study was set at 0.05.

## 3. Results

### 3.1. Descriptive Analysis of the Sample

Table 1 shows the sociodemographic and legal–criminal characterization of the sample. The mean age of the participants was 40.61 years (SD = 10.91), ranging from 20 to 69 years. Regarding educational qualifications, more than half of the participants (54.3%) had completed the minimum mandatory education (12 years), and 28.7% had completed basic education (9 years). Concerning substance use history, we observed that the majority of participants (56.4%) reported no substance use. In terms of parenthood, it is possible to observe that most women were mothers (*n* = 76; 80.9%), and regarding familial support, most participants reported having it (79.8%).

Regarding the crime for which they were sentenced, it is possible to conclude that some participants were serving sentences for multiple crimes simultaneously, so we chose to dichotomize these variables (i.e., coding for whether participants had committed each specific crime or not). The most common crime in our sample was drug trafficking (87.2%), followed by theft (40.4%), homicide (34%), and robbery (25.5%). These findings align with annual statistics, which, according to the Annual Internal Security Report ([85]), shows that drug trafficking remains one of the most common crimes committed in the country, being the most common crime committed or allegedly committed by women in the prison system in the country (26%) ([30]). Similarly, property crimes, particularly theft, continue to be highly representative of reported criminality, accounting for about 26.3% of all reported crimes ([85]).

#### 3.1.1. Analysis of Self-Control Levels in the Sample

To understand the self-control levels presented by the sample, we conducted a descriptive statistical analysis of the scores obtained on the Self-Control Scale. As shown in Table 2, the mean self-control score was 31.50 (*SD* = 7.90), with a minimum score of 16 points and a maximum of 57. Since higher scores indicate greater levels of self-control, we can infer that the sample in question demonstrates average levels of self-control ([19]).

#### 3.1.2. Analysis of Self-Control Levels Based on Substance Use History

Regarding self-control (Table 3), the results indicate significant differences between participants with a history of substance use and those without. Participants without a history of substance use displayed higher levels of self-control compared to those with a history of use (U = 790; *p* = 0.024; r = 0.233).

#### 3.1.3. Analysis of Self-Control Levels Based on the Commission of Violent Crimes

Table 4 shows that, regarding self-control, there are statistically significant differences between participants who committed homicide and those who did not (U = 708.5; *p* = 0.024; r = 0.233). Women who committed homicide exhibited higher levels of self-control compared to those who were not convicted of this crime.

However, concerning robbery, the results indicate no statistically significant differences in self-control levels between women who committed this crime and those who did not (U = 602.5; *p* = 0.060; r = 0.122). Despite this, women who did not report committing robbery showed slightly higher self-control levels.

#### 3.1.4. Analysis of Psychopathy Levels in the Sample

As shown in Table 5, the mean psychopathy score on the LSRP-VP scale for the sample was 66.36 (*SD* = 8.51). This indicates significant levels of psychopathy, as the scores were above the reference mean (*M* = 48.98). Additionally, the results reveal that the mean scores for primary psychopathy—associated with traits such as selfishness, carelessness, and manipulation—were significantly higher than those for secondary psychopathy, which is linked to more impulsive and self-destructive characteristics ([17]).

For Factor 1 (primary psychopathy), the sample’s mean score was 40.34 (*SD* = 5.95), notably above the reference mean (*M* = 28.90; *SD* = 6.20). For Factor 2 (secondary psychopathy), the mean score was 26.02 (*SD* = 4.06), which, while significantly above the reference mean (*M* = 20.08; *SD* = 5.22), was the lower of the two factors, which suggests that participants in this sample scored less on traits related to impulsivity and self-destructive behaviors (secondary psychopathy), even though they exceeded the reference mean.

#### 3.1.5. Analysis of Psychopathy Levels Based on Substance Use History

In terms of psychopathy, no statistically significant differences were found between participants with and without a history of substance use, both in the total scale (U = 1019.5; p = 0.609; r = 0.052) or in Factor 1 (U = 1063.5; p = 0.861; r = 0.018) and Factor 2 (U = 979.5; *p* = 0.413; r = 0.084). Nonetheless, we observe that participants reporting a history of psychoactive substance use show higher average levels of psychopathy compared to those who reported no such history. This pattern is repeated for both primary psychopathy (Factor 1) and secondary psychopathy (Factor 2), with the latter emphasizing psychopathic traits more strongly (Table 6).

#### 3.1.6. Analysis of Psychopathy Levels Based on the Commission of Violent Crimes

Regarding the commission or non-commission of violent crimes, it is possible to observe that for the crime of homicide, no statistically significant differences were found in the total psychopathy scale (U = 790.5; *p* = 0.107; r = 0.166) or in Factor 1 (U = 897; *p* = 0.448; r = 0.078). However, for Factor 2, significant differences exist between participants who committed this type of crime and those who did not (U = 737.5; *p* = 0.042; r = 0.210), with the latter showing higher levels of psychopathy.

Additionally, no statistically significant differences were found between women who committed robbery and those who did not, either on the total psychopathy scale or its factors (Table 7). Nevertheless, it is worth noting that participants who reported committing this crime presented higher average psychopathy scores (*M* = 55.04) compared to those who reported not committing this crime (*M* = 45.06).

#### 3.1.7. Analysis of the Relationship Between Self-Control and Psychopathy Based on the Commission of Violent Crimes

As shown in Table 8, there is a negative and statistically significant correlation between self-control and total psychopathy (Rpb = −0.335; *p* < 0.001), indicating that in the current sample, lower levels of self-control are associated with higher levels of psychopathy.

Regarding primary psychopathy (Factor 1), there is also a negative correlation with self-control, although it is not statistically significant (Rpb = −0.183; *p* = 0.077). For secondary psychopathy (Factor 2), a negative and significant correlation with self-control was found (Rpb = −0.419; *p* < 0.001), suggesting that lower levels of self-control are associated with higher secondary psychopathy traits (Factor 2).

It is also evident that both self-control (Rpb = −0.022; *p* = 0.832) and psychopathy (Rpb = −0.006; *p* = 0.952) are negatively associated with the commission of violent crimes. This means that lower levels of these constructs are linked to a greater likelihood of violent crime involvement. However, in neither case is this relationship statistically significant.

#### 3.1.8. Investigating the Relationship Between Self-Control and Psychopathy: A Predictive Analysis

To gain a deeper understanding of the relationship between self-control and psychopathy, specifically whether psychopathy is a significant predictor of self-control, a simple linear regression analysis was conducted (Table 9). The regression model proved to be statistically significant (F = 11.65; *p* < 0.001; R^2^ = 0.112). This indicates that psychopathy, within this sample, has a predictive effect on self-control, although it accounts for only 11.2% of the observed variance.

## 4. Discussion

This study highlights the scarcity of research on psychological constructs in incarcerated women. The primary aim of the present study was to investigate levels of self-control and psychopathy in a female sample in Portugal and explore their relationship with substance use and violent offenses (homicide and robbery).

Firstly, the self-control scale scores indicate that the sample presents average levels of self-control. These findings reveal a certain duality: on one hand, it was expected that self-control levels would be lower, given that the sample consisted of incarcerated individuals, who, according to the literature, tend to exhibit lower levels of self-control ([59]; [74]). On the other hand, the results align with expectations, or even exceed them, as the participants were women, and the literature suggests that female samples generally exhibit relatively higher levels of self-control ([81]). These initial results underscore the importance of considering the gender variable when applying this psychological construct to prison populations as a way of allowing us to assess the extent to which existing literature-based inferences about the relationship between self-control and incarcerated populations are applicable to women.

Furthermore, consistent with prior studies ([47]), participants with no history of substance use showed higher self-control levels than those with such history. This may be explained by adverse childhood experiences. Studies indicate that, for women, substance use is often associated with traumatic experiences, such as domestic violence or sexual abuse, which contribute to the initiation and maintenance of substance use ([71]), and can lead to structural changes in the brain, which, in turn, may result in lower levels of self-control ([53], as cited by [15]). In fact, incarcerated populations have a higher prevalence of adverse childhood experiences compared to the general community ([43]), particularly evident in the context of childhood sexual abuse, which is also significantly more prevalent among incarcerated women than in the general population ([46]).

Contrary to expectations that violent offenders would show lower self-control ([76]), this was not observed among women convicted of homicide. A likely explanation is that this crime may have been committed, by some women, in response to prolonged domestic violence. Research shows that many women who commit homicide against their partners do so as an act of self-defense, survival, or to escape the violence they endure ([12]; [50]; [55]; [68]; [92]). Therefore, this sample may include women whose motivations for committing homicide are rooted not in a lack of self-control but in the need to survive or escape abusive contexts.

Regarding robbery, no significant differences in self-control levels were found between women who committed this crime and those who did not. However, unlike homicide, women who reported committing robbery exhibited lower self-control levels, aligning with expectations. Despite this, it is important to emphasize that while self-control is a significant factor in understanding criminal behavior, it does not act in isolation, and its relationship with specific crimes has not been sufficiently and satisfactorily explored ([94]).

Regarding psychopathy, the prevalence of this construct in the sample indicates significant psychopathic traits, with values notably above the average on both the total scale and its factors. In fact, there is speculation that underdiagnosis of psychopathy in incarcerated women may occur due to gender biases or sampling errors in research, potentially leading to unrealistically low estimates of this disorder in women ([79]). Associated with this issue, there is the fact that women are sometimes inappropriately diagnosed with borderline personality disorder—a disorder substantially prevalent in incarcerated populations, along with antisocial personality disorder ([7]; [8]). Thus, the levels of psychopathy observed in this sample challenge the conventional notion that women inherently exhibit lower psychopathy levels, which may reflect gender biases and methodological limitations as outlined above. The results from this study highlight the importance of further examining this construct in female populations and incarcerated women specifically ([39]). Future studies should also explore whether significant associations exist between psychopathy and non-violent crimes.

Additionally, we alert that Cronbach’s alpha values obtained for Factors 1 and 2 on the LSRP-VP scale, particularly Factor 2, were notably low. Therefore, interpretations of these results should be cautious.

Women with a history of substance use scored higher on overall psychopathy, particularly on secondary psychopathy, aligning with literature linking psychopathic traits to increased substance use ([49]; [86]). The LRSP-VP scale reflects the two-factor structure of the PCL-R ([45]), with “secondary psychopaths” associated with lower behavioral inhibition and greater substance use, including alcohol, cocaine, and marijuana ([49]). This pattern may be explained by adverse childhood experiences (ACEs), which are strongly linked to both substance use and traits of secondary psychopathy([29]; [61]; [66]). Thus, it is plausible that early traumatic experiences, which are much more prevalent in incarcerated women than in the community ([46]), contribute to both heightened secondary psychopathic traits and the adoption of maladaptive coping mechanisms such as substance use, emphasizing the complex interplay between trauma, psychopathy, and behavioral outcomes in female forensic populations. Despite its significance, the link between psychopathy and substance use in female offenders remains underexplored ([32]).

The relationship between substance abuse and crime has been emphasized in various forensic contexts ([36]).However, it is important to note that not all substance use is inherently maladaptive. Illicit substance use is common, and some individuals, even when fully aware of the associated risks and benefits, choose to engage in it ([20]). That being said, moderate or low levels of substance use may not necessarily be harmful and could even have beneficial effects under certain conditions ([10]), with this category of users being referred to as “non-problematic consumers” ([20]). Thus, although the literature establishes a link between psychopathy, self-control, and increased substance use, it is essential to consider that such consumption may not always be problematic, whereby we consider this to be pertinent in the interpretation of our findings.

Regarding violent crimes, we anticipated significant differences in psychopathy levels between women who committed homicide and those who did not. However, no such differences were observed. Women who committed homicide exhibited lower psychopathy levels compared to the reference mean (*M* = 48.98). This finding aligns with a similar study ([56]), which found no greater prevalence of psychopathy among individuals who had committed violent offenses, contrasting with the broader literature. These results suggest that the relationship between psychopathy and homicide perpetrated by women may not always be clear and linear, with some individuals who commit such crimes lacking psychopathic traits entirely. Another potential explanation for this discrepancy is the over-reliance on male-dominated samples in psychopathy research. Most studies investigating psychopathy are developed with predominantly male populations, leading to assumptions that male-based findings are “transferable” to female populations, possibly bringing substantial consequences ([95]). This gendered lens may overlook critical nuances, like the before-mentioned possibility that women convicted of homicide often commit such acts as a response to prolonged abuse or as a survival mechanism, rather than as a result of psychopathic tendencies.

Regarding Factor 1 of psychopathy, no statistically significant differences were found between women who committed homicide and those who did not. Conversely, significant differences emerged in Factor 2. Although Factor 1 lacked statistical power in terms of homicide, it showed higher average scores than Factor 2. Concerning the lower internal consistencies of both factors, caution is needed when interpreting our findings. It may be necessary to revise this instrument to improve its reliability, as it might not be the most adequate one to administer to incarcerated populations, particularly incarcerated women. Research shows that it is crucial to use instruments that take into account the unique life experiences and contextual factors of incarcerated individuals in order to effectively assess them ([52]). Given this, it is possible that the observed low internal consistency may be attributed to how participants interpreted the items (i.e., the language used in the assessment tool may be too complex for them). This could have affected their responses and, consequently, compromised the tool’s internal validity.

Furthermore, regarding the association between self-control, psychopathy, and violent crimes, we obtained results indicating that the lower the levels of self-control, the higher the psychopathic traits, confirming the initially established hypothesis. It is widely known that for individuals with psychopathy, difficulty in self-control is a very common characteristic, potentially leading them to react disproportionately to a stimulus ([58]). Thus, these results align with the existing literature that shows that lower self-control is significantly associated with higher psychopathic traits ([51]; [65]). However, neither self-control nor psychopathy were significantly correlated with the commission of violent crimes.

Regarding self-control, our results contradict what the literature has shown over time. The study by [26] ([26]) demonstrated a significant association between low levels of self-control and the commission of violent offenses. Another study also emphasized self-control as a significant predictor of criminal behavior ([77]). This leads us to believe that, as mentioned earlier, the participants in this sample may have motivations that go beyond this construct, thereby self-control not constituting as a significant predictor of violent crime.

Regarding psychopathy, our results align with those obtained in the study conducted by [26] ([26]), where psychopathy was not significantly associated with the commission of violent crimes in any of the analyses performed. However, there is a wide range of studies that reveal psychopathy as a strong predictor of violent crimes ([6]; [42]; [48]). In our sample, regarding psychopathy, the situation mirrors that of self-control, as it does not constitute a significant predictor of violent crime.

As mentioned, self-control and psychopathy were significantly correlated with each other. When exploring this relationship further, we observed that psychopathy emerged as a significant predictor of self-control levels in this sample. In fact, research has shown that low self-control is one of the inherent characteristics of psychopathy ([78]); however, the literature has not extensively explored this predictive relationship. In this sense, we consider it relevant to investigate this relationship in future studies.

## 5. Limitations

Despite the contributions of this study, it is not immune to limitations, which we acknowledge may have impacted the results. One such limitation is the small sample size, limiting the generalizability of the results. The use of self-report scales also inherently introduces the possibility of bias. While self-report scales are useful, they have disadvantages, particularly in forensic populations, where social desirability may influence responses ([73]). Furthermore, questionnaires were sometimes applied in less-than-ideal contexts, such as cafeterias or libraries, where multiple interruptions occurred, potentially having a counterproductive effect on participants’ responses. Therefore, we suggest replicating this study with a larger sample and triangulating data collection methods to achieve potentially more representative and reliable results.

## 6. Conclusions

The present study addressed critical gaps in the understanding of female criminal behavior, particularly given the scarcity of research exploring psychological constructs within this population and the specificities that distinguish it from male offending. The primary objective was to examine the manifestation of self-control and psychopathy among incarcerated women, and to analyze their relationship with sociodemographic variables and the commission of violent crimes, specifically homicide and robbery, which provided us with highly insightful findings.

We recommend further research on self-control and psychopathy, to better understand the differences between genders in these constructs and the extent to which these differences persist Also, the relationship between these variables in women and recidivism requires continuous investigation, as they are crucial to understanding possible differences in female criminal behavior, which is essential to developing more targeted and effective interventions.

Although not the main focus of this investigation, we also suggest continuing the study of female crime, specifically homicide, qualitatively. To gain richer and more comprehensive insights into the motivations behind crimes committed by women, it is insufficient to merely evaluate their prevalence; it is also essential to examine all the nuances and specificities surrounding these crimes.

Finally, despite the aforementioned limitations, the relevance of this research is recognized, both in relation to the selected sample and the constructs used. This is particularly significant given the scarcity of studies that combine these two aspects, the intersection of psychological constructs and female offending. We successfully achieved the objective of exploring these variables within a female population, even though it raised additional questions, an outcome that is, in itself, valuable and informative. More than providing answers, we aimed to contribute to the literature on women who have committed crimes. The results obtained serve as a starting point for future research that can further illuminate the psychological factor influencing female criminal behavior, particularly within prison populations.

## 7. Implications for Practice

The findings in this study highlight the importance of a gender-sensitive approach when conducting research in the prison context. Addressing and researching the manifestation in women of extensively investigated variables in men is a crucial starting point for future program and intervention conception, in order to make these as effective as possible. Gender-responsive practices are indispensable for the rehabilitation of incarcerated women ([2]), and therefore, it is crucial to conduct gender-sensitive assessments to ensure that programs and interventions are appropriately tailored to their specific needs. As such, this study emphasizes that these well-established psychological constructs and their relationship with criminal behavior may be less clear in female samples, highlighting the need to approach these issues through a gender-sensitive lens.

## Figures and Tables

**Table 1 behavsci-15-00656-t001:** Sociodemographic characterization.

	N (%)
** *Education* **	
12 years	51 (54.3)
9 years	27 (28.7)
Illiterate	2 (2.1)
** *Marital Status* **	
Single	42 (44.7)
Married	19 (20.2)
Divorced/Separated	28 (29.8)
Widowed	5 (5.3)
** *Parenthood* **	
Children	76 (80.9)
No Children	18 (19.1)
** *Familial Support* **	
Yes	75 (79.8)
No	19 (20.2)
** *Substance Use History* **	
Yes	41 (43.6)
No	53 (56.4)

**Table 2 behavsci-15-00656-t002:** Self-control scores.

	N	Min–Max	M (*SD*)
**Total BSCS ***	94	[16–57]	31.50 (7.90)

* Basic Self-Control Scale.

**Table 3 behavsci-15-00656-t003:** Self-control scores based on substance use (Mann–Whitney U Test).

	Substance Use	N	M	U
**Self-Control**	Yes	41	40.27	790 *
	No	53	53.09	

Note: * *p* < 0.05.

**Table 4 behavsci-15-00656-t004:** Self-control scores based on the commission of violent crimes (Mann–Whitney U Test).

		N	M	U
	**Homicide**			
	Yes	32	56.36	708.5 *
	No	62	42.93	
**Self-Control**				
	**Robbery**			
	Yes	23	38.20	602.5
	No	71	50.51	

Note: * *p* < 0.05.

**Table 5 behavsci-15-00656-t005:** Psychopathy scores.

	N	Min–Max	M (*SD*)
**Total LSRP-VP ***	94	[45–88]	66.36 (8.51)
**Factor 1**	94	[27–57]	40.34 (5.95)
**Factor 2**	94	[18–37]	26.02 (3.98)

* Levenson’s Self-Report Psychopathy Scale—Portuguese Version.

**Table 6 behavsci-15-00656-t006:** Psychopathy scores based on substance use (Mann–Whitney U Test).

	Substance Use	N	M	U
**Psychopathy**	Yes	41	49.13	1019.5
	No	53	46.24	
**Primary Psychopathy**	Yes	41	48.06	1063.5
	No	53	47.07	
**Secondary Psychoathy**	Yes	41	50.11	979.5
	No	53	45.48	

**Table 7 behavsci-15-00656-t007:** Psychopathy scores based on the commission of violent crimes (Mann–Whitney U Test).

		N	M	U
**Psychopathy**	**Homicide**			
	Yes	32	41.20	790.5
	No	62	50.75	
	**Robbery**			
	Yes	23	55.04	643
	No	71	45.06	
**Factor 1**	**Homicide**			
	Yes	32	44.53	897
	No	62	49.03	
	**Robbery**			
	Yes	23	55.35	636
	No	71	44.96	
**Factor 2**	**Homicide**			
	Yes	32	39.55	737.5 *
	No	62	51.60	
	**Robbery**			
	Yes	23	53.02	689.5
	No	71	45.71	

Note: * *p* < 0.05.

**Table 8 behavsci-15-00656-t008:** Relation between self-control and psychopathy, and the commission of violent crimes (Point-Biserial Correlation).

	Self-Control	Psychopathy	Factor 1	Factor 2	Violent Crimes
	R*pb*	R*pb*	R*pb*	R*pb*	R*pb*
**Self-Control**	1				
**Psychopathy**	−0.355 **	1			
**Factor 1**	−0.183	0.905 **	1		
**Factor 2**	−0.419 **	0.780 **	0.451 **	1	
**Violent Crimes**	−0.022	−0.006	0.054	−0.114	1

Note: ** *p* < 0.001.

**Table 9 behavsci-15-00656-t009:** Linear regression between self-control and psychopathy (simple linear regression).

			Self-Control	
	R^2^	β	*t*	*p*
**Psychopathy**				
	0.112	−0.311	−3.414	<0.001

## Data Availability

The raw data supporting the conclusions of this article will be made available by the authors on request.

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
