# Peer review of "Violence Under Control: Self-Control and Psychopathy in Women Convicted of Violent Crimes"

_behavsci, 2025, doi:10.3390/bs15050656_

Round 1

Reviewer 1 Report

Comments and Suggestions for Authors

The manuscript presents research findings on levels of self-control and psychopathy among women incarcerated in Portuguese prisons, with emphasis on the relationship between these variables and substance use and violent crimes. Its main contribution is its focus on female offender populations, a group that has received relatively limited research attention in the field.

Literature Review - The literature review is comprehensive and grounded in both foundational criminological theory and recent empirical studies. The authors appropriately highlight the historical reliance on male-focused diagnostic criteria in psychopathy research and criminal typologies, making a strong case for the importance of gender-sensitive frameworks. This section contributes well to positioning the study within a significant gap in the field.

However, the authors note that "the incarceration rate of women in Portugal, compared to other European countries, is quite expressive" (p. 3, last paragraph, line 146 – 147) - this sentence is unclear and should be clarified. In addition, it is appropriate to add data on the incarceration rates of women and men in Portugal, the characteristics of prisoners and the types of crime common among them.

Methodology – The study used a sample of 94 women in prison in Portugal, with validated instruments. The authors noted several significant methodological limitations, relating to the reliability of the instruments (low Cronbach's alpha values ​​for psychopathy factors (0.60 and 0.53), a relatively small sample size (n=94), and data collection under non-ideal conditions. These are serious limitations that impair the weighing of the findings, as the authors also noted.

However, there is another limitation that affects the validity of the findings, and it is the lack of distinction between types of murder: the study does not distinguish between different types of murder, motives, and the participants' history as victims of violence. This is a fundamental limitation that should be addressed and perhaps even attempted to be remedied.

Results - The results are clearly and systematically presented. The authors demonstrate commendable transparency, reporting both significant and non-significant findings without overstating the implications. The statistical analyses are appropriate and show that the relationships between psychopathy, self-control, and violent offending are more complex than commonly assumed.

Particularly noteworthy is the finding that women convicted of murder scored higher on self-control—this result is intriguing and counterintuitive. However, it would benefit from deeper theoretical exploration. The possible explanation—that such offenses may stem from prolonged victimization or self-defense—is mentioned but not thoroughly developed.

Discussion – The discussion section reflects thoughtful engagement with the findings and is characterized by a cautious and theoretically informed approach. The authors commendably attempt to situate their results within a broader framework that includes gender dynamics, trauma history, and critiques of traditional criminological theory. Particularly compelling is the effort to reconsider the association between self-control and violent offending among women, especially in light of the counterintuitive finding that women convicted of murder exhibited higher levels of self-control.

However, a major limitation of the discussion lies in the absence of any classification or contextualization of the homicides committed. The study does not provide information on the nature of the murders—whether they involved intimate partners, children, family members, or strangers. This omission significantly constrains the interpretive power of the results, particularly regarding psychological motivation and behavioral dynamics. A distinction between instrumental homicides (e.g., defensive actions, protective responses, escape from abuse) and reactive/emotional killings would allow for a more nuanced and meaningful discussion of self-control as a contributing factor.

Conclusion and Recommendation – While the manuscript conclusions are well presented, there is no reference to the practical implications of the research. It is advisable to add references to therapeutic, legal, incarceration policy, prevention programs, and professional training implications. A significant expansion of the implications section would have strengthened the article and increased its relevance for professionals.

Author Response

Comment 1: 

“However, the authors note that "the incarceration rate of women in Portugal, compared to other European countries, is quite expressive" (p. 3, last paragraph, line 146 – 147) - this sentence is unclear and should be clarified. In addition, it is appropriate to add data on the incarceration rates of women and men in Portugal, the characteristics of prisoners and the types of crime common among them.”

Answer 1: Thank you for this input. We agree that the sentence sounded unclear and we have now clarified it. We have also added some data on the incarceration rates of men and women in Portugal; however, we tried not to extend it too much to avoid making it sound too comparative. 

Commment 2: 

“However, there is another limitation that affects the validity of the findings, and it is the lack of distinction between types of murder: the study does not distinguish between different types of murder, motives, and the participants' history as victims of violence. This is a fundamental limitation that should be addressed and perhaps even attempted to be remedied.”

Answer 2: Thank you for your comment. We realize this crime is committed in different circumstances and ideally must be analyzed through this lens, considering all its specificities. However, no thorough qualitative information on the commission of this crime was available to us. As such, we have made a statement that this was done so. 

Comment 3: “Particularly noteworthy is the finding that women convicted of murder scored higher on self-control—this result is intriguing and counterintuitive. However, it would benefit from deeper theoretical exploration. The possible explanation—that such offenses may stem from prolonged victimization or self-defense—is mentioned but not thoroughly developed.”

Answer 3: Thank you. We have developed this argument and added more references corroborating this idea. 

Comment 4: 

“However, a major limitation of the discussion lies in the absence of any classification or contextualization of the homicides committed. The study does not provide information on the nature of the murders—whether they involved intimate partners, children, family members, or strangers. This omission significantly constrains the interpretive power of the results, particularly regarding psychological motivation and behavioral dynamics. A distinction between instrumental homicides (e.g., defensive actions, protective responses, escape from abuse) and reactive/emotional killings would allow for a more nuanced and meaningful discussion of self-control as a contributing factor.”

Answer 4: Thank you. We agree that this suggestion makes a lot of sense and would allow for a deeper analysis of our findings. However, no information on the circumstances of these homicides were available to us.

Comment 5: 

“While the manuscript conclusions are well presented, there is no reference to the practical implications of the research. It is advisable to add references to therapeutic, legal, incarceration policy, prevention programs, and professional training implications. A significant expansion of the implications section would have strengthened the article and increased its relevance for professionals.”

Answer 5: Thank you for this last observation. We have now added a “Implications for Practice” section. 

Reviewer 2 Report

Comments and Suggestions for Authors

I thank the authors for the opportunity to review this manuscript. The highly interesting study addresses a topic still underrepresented in the current literature. Below are a few suggestions to enhance the paper's overall quality and clarity.

The introduction is certainly engaging, but I believe it requires revision of the length and paragraph structure. A clearer deductive approach should be maintained throughout, as this is occasionally lost in the current version. The authors have chosen to begin the paragraph subdivision of the introduction, starting with the section on self-control. However, since this study involves a small, all-female forensic sample, it would be more appropriate to introduce this specific context first. In this regard, I recommend considering the following study: DOI: 10.1080/09540261.2024.2378070

I suggest a general revision of the entire introduction to make it more concise and to strengthen the logical flow between sections, avoiding excessive digressions. Furthermore, the research gaps this study intends to address are not clearly articulated. Lines 169–187, including Table 1, would be better placed in the Results section.

I am also curious as to why the PCL-R was not used. Could the authors clarify the rationale behind the choice of the specific scale employed?

I recommend avoiding excessive detail in the description of the psychometric tools. Instead, briefly define the outcome measures, the factors assessed, the validation status of the instrument in Portuguese, the administration procedures, the scoring system, and the reasons supporting its selection.

The Results section should begin with descriptive statistics.

None of the tables are self-explanatory. Please indicate which statistical tests were used, and always specify the meaning of acronyms in the footnotes. There are also some formatting issues—for instance, see Table 9.

The Discussion section is overly lengthy. A separate paragraph dedicated to the study’s limitations is needed, as these should not be included in the Conclusions. Proposals for future research should be more clearly articulated, and the Conclusions themselves should be made more impactful.

Author Response

Comment 1: 

“The introduction is certainly engaging, but I believe it requires revision of the length and paragraph structure. A clearer deductive approach should be maintained throughout, as this is occasionally lost in the current version. The authors have chosen to begin the paragraph subdivision of the introduction, starting with the section on self-control. However, since this study involves a small, all-female forensic sample, it would be more appropriate to introduce this specific context first. In this regard, I recommend considering the following study: DOI: 10.1080/09540261.2024.2378070."

Answer 1: Thank you for this observation. We have altered the introduction to make it more concise, and we have added the introduction on this specific context. Unfortunately, the suggested reference was not available to us. 

Comment 2: 

“Furthermore, the research gaps this study intends to address are not clearly articulated. Lines 169–187, including Table 1, would be better placed in the Results section.”

Answer 2: This information has been placed in the Results section. Thank you for your input on this. 

Comment 3: 

“I am also curious as to why the PCL-R was not used. Could the authors clarify the rationale behind the choice of the specific scale employed?”

Answer 3: Thank you for this curiosity, it is very pertinent. The PCL-R was not used because it is not a self-report instrument. In this specific context, it would not be feasible to administer the PCL-R to all participants, as it would be lengthy. Therefore, we opted for the LRSP tool, so that participants could answer it by themselves. 

Comment 4: 

“I recommend avoiding excessive detail in the description of the psychometric tools. Instead, briefly define the outcome measures, the factors assessed, the validation status of the instrument in Portuguese, the administration procedures, the scoring system, and the reasons supporting its selection.”

Answer 4: Changes have been made to this section to make it briefer. Thank you.

Comment 5: 

“The Results section should begin with descriptive statistics.”

Answer 5: We have changed this. Thank you. 

Comment 6: 

“None of the tables are self-explanatory. Please indicate which statistical tests were used, and always specify the meaning of acronyms in the footnotes. There are also some formatting issues—for instance, see Table 9.”

Answer 6: This has been corrected. Thank you for this observation.

Comment 7: 

“The Discussion section is overly lengthy. A separate paragraph dedicated to the study’s limitations is needed, as these should not be included in the Conclusions. Proposals for future research should be more clearly articulated, and the Conclusions themselves should be made more impactful.”

Answer 7: Thank you. We have revised the discussion section and made it as concise as possible while trying not to lose important details. Likewise, we have made some changes in the conclusion section. 

Reviewer 3 Report

Comments and Suggestions for Authors

The paper is well organized and written. It is also highly interesting. Two major criticisms include 1. Not providing the items in the two scales (p. 5) and 2. The paper is tautological. The definition of APD includes crime throughout so of course there is a relationship between the two concepts. It is true by definition (p. 2). Also you're wrong on the claim that most people with APD are incarcerated (p. 3). They're actually in big business (Hare). Finally you can't say psychopathy predicts social control; social control is also part of the definition (p. 9).

Author Response

Comment 1: 

“The paper is well organized and written. It is also highly interesting. Two major criticisms include 1. Not providing the items in the two scales (p. 5) and 2. The paper is tautological. The definition of APD includes crime throughout so of course there is a relationship between the two concepts. It is true by definition (p. 2). Also you're wrong on the claim that most people with APD are incarcerated (p. 3). They're actually in big business (Hare). Finally you can't say psychopathy predicts social control; social control is also part of the definition (p. 9).”

Answer: Thank you for your feedback. In terms of the scale items, we have asked for permission to use the tools, but not to disclose it to the public. So, we will not be able to share that information in this article. Regarding the second issue raised, like we mentioned in the article introduction, we do not believe an APD diagnosis necessarily means the person will engage in criminal behavior, arguing this relationship is not causal (p. 3). Regarding the third observation, we do agree that it is not true that most people with APD are incarcerated, so we altered the text. Thank you.

Reviewer 4 Report

Comments and Suggestions for Authors

First of all, I want to acknowledge the importance of this study. Investigating psychopathy and self-control in incarcerated women is no small task—methodologically, ethically, or conceptually. The fact that you chose to focus on this population, and to engage seriously with gender-specific pathways to violent behavior, makes this paper a meaningful and much-needed contribution to the literature. well done.

That said, there are -  several points - that need to be strengthened before the paper can reach its full potential.

1. Psychopathy: You use the LSRP and its two-factor structure, which is appropriate. However, the distinction between primary and secondary psychopathy isn’t well developed in the theoretical section. More importantly, the gendered manifestation of these traits—especially how secondary psychopathy might intersect with trauma and impulsivity in women—is not sufficiently explored. I suggest you briefly expand on this early on, and also revisit it in the discussion when interpreting your results.

2. Measurement issues -The Cronbach’s alpha for is quite low, which raises some questions about construct validity. You do mention this, which is good, but I’d encourage you to take it a step further. Maybe discuss possible reasons for this and what it might mean for interpretation. This kind of transparency will only strengthen your work.

3. Unexpected findings deserve more than intuition - One of your most striking findings is that women who committed homicide scored higher on self-control. You offer a plausible explanation (i.e., violence as a reaction to prolonged abuse or a survival strategy), but this remains speculative without stronger support. I’d urge you to bring in literature from victimology that specifically addresses this pattern. There’s a rich body of work here.

4. The discussion section is thoughtful, but  - slightly  - repetitive. There are moments in the discussion where ideas are reiterated without adding much depth. Consider trimming redundancies and sharpening your takeaways. A tighter discussion will make your core contributions stand out more.

5. Statistical depth: I appreciate the caution you took in using non-parametric tests and explaining the limits of your data. (this part is overall very well) .That said, if you're already testing for predictive relationships, why not go a bit further? It’s not essential, but it would strengthen the quantitative contribution.

6. Practical implications: Even just a few lines about what these findings suggest for practice—whether in prison programming, risk assessment, or treatment planning—would help ground the research

Author Response

Comment 1: 

“Psychopathy: You use the LSRP and its two-factor structure, which is appropriate. However, the distinction between primary and secondary psychopathy isn’t well developed in the theoretical section. More importantly, the gendered manifestation of these traits—especially how secondary psychopathy might intersect with trauma and impulsivity in women—is not sufficiently explored. I suggest you briefly expand on this early on, and also revisit it in the discussion when interpreting your results.”

Answer 1: We have now added a section in the introduction explaining the difference between the factors, and a new point in the discussion on the secondary psychopathy. Thank you.

Comment 2: 

“Measurement issues -The Cronbach’s alpha for is quite low, which raises some questions about construct validity. You do mention this, which is good, but I’d encourage you to take it a step further. Maybe discuss possible reasons for this and what it might mean for interpretation. This kind of transparency will only strengthen your work.”

Answer 2: We have added some discussion on this. Thank you for your valuable input!

Comment 3: 

“Unexpected findings deserve more than intuition - One of your most striking findings is that women who committed homicide scored higher on self-control. You offer a plausible explanation (i.e., violence as a reaction to prolonged abuse or a survival strategy), but this remains speculative without stronger support. I’d urge you to bring in literature from victimology that specifically addresses this pattern. There’s a rich body of work here.”

Answer 3: We have added literature that addresses this issue. Thank you for this observation.

Comment 4: 

“The discussion section is thoughtful, but - slightly - repetitive. There are moments in the discussion where ideas are reiterated without adding much depth. Consider trimming redundancies and sharpening your takeaways. A tighter discussion will make your core contributions stand out more.”

Answer 4:We have tried to make the discussion concise to make it less repetitive, without losing its detail. Thank you for this feedback.

Comment 5: 

“Statistical depth: I appreciate the caution you took in using non-parametric tests and explaining the limits of your data. (this part is overall very well). That said, if you're already testing for predictive relationships, why not go a bit further? It’s not essential, but it would strengthen the quantitative contribution.”

Answer 5: We thank you for this suggestion, and we agree it makes a lot of sense. Although acknowledging that other analytical plans could be applied, this option, in our view, adequately responds to what we defined as the research question. In addition, additional tests would imply an increase of aspects in the discussion, which already had to be shortened to address other suggestions and would not be feasible due to space limitations, and even the deadline for this review. 

Comment 6: 

“Practical implications: Even just a few lines about what these findings suggest for practice—whether in prison programming, risk assessment, or treatment planning—would help ground the research.”

Answer 6: We have added a “Implications for Practice” section. Thank you for this pertinent suggestion.

Round 2

Reviewer 2 Report

Comments and Suggestions for Authors

I appreciate the authors' revisions. Most of the suggestions provided in the first round of review have been adequately addressed, resulting in a more precise and structured manuscript.
Nevertheless, a significant concern remains regarding the omission of the suggested reference, which is directly pertinent to the context of forensic female populations. The authors' justification, stating that the reference was "not available," is not sufficient within the standards of scholarly work. If the reference was inaccessible, alternative solutions (e.g., contacting the publisher and consulting institutional libraries) should have been explored.
Although the Discussion section has been condensed, I encourage the authors to refine further and synthesize the content to enhance clarity and focus.
I believe that with these minor revisions, the manuscript will be suitable for publication.

Author Response

Comment 1: 

“I appreciate the authors' revisions. Most of the suggestions provided in the first round of review have been adequately addressed, resulting in a more precise and structured manuscript.
Nevertheless, a significant concern remains regarding the omission of the suggested reference, which is directly pertinent to the context of forensic female populations. The authors' justification, stating that the reference was "not available," is not sufficient within the standards of scholarly work. If the reference was inaccessible, alternative solutions (e.g., contacting the publisher and consulting institutional libraries) should have been explored.
Although the Discussion section has been condensed, I encourage the authors to refine further and synthesize the content to enhance clarity and focus. I believe that with these minor revisions, the manuscript will be suitable for publication.”

Answer 1: Thank you for your valuable feedback! The discussion has been further condensed as much as possible, while ensuring the preservation of key contributions and interpretations. Thank you also for the suggestion regarding the inclusion of the paper by Buongiorno et al., 2024. We have carefully considered and analyzed it, and while we appreciate the relevance of this work within the context of forensic psychiatric populations, we believe it does not fully align with the specific focus of this paper, which is centered on incarcerated women in general prison settings. Moreover, the suggested reference does not address the main psychological variables examined in this study – namely, psychopathy and self-control – which are central to our analysis. As the journal’s Guidelines mention, if we believe a suggested reference does not entirely fit in the scope of the paper, we should not accept it. However, we appreciate the suggestion and will consider this reference for future studies with a forensic psychiatric population. 

Reviewer 3 Report

Comments and Suggestions for Authors

None

Author Response

Thank you.